# Spatiotemporal Graph Guided Multi-modal Network for Livestreaming Product Retrieval

## ABSTRACT

With the rapid expansion of e-commerce, more consumers have become accustomed to making purchases via livestreaming. Accurately identifying the products being sold by salespeople, i.e., livestreaming product retrieval (LPR), poses a fundamental and daunting challenge. The LPR task encompasses three primary dilemmas in real-world scenarios: 1) the recognition of intended products from distractor products present in the background; 2) the video-image heterogeneity that the appearance of products showcased in live streams often deviates substantially from standardized product images in stores; 3) there are numerous confusing products with subtle visual nuances in the shop. To tackle these challenges, we propose the Spatiotemporal Graphing Multi-modal Network (SGMN). First, we employ a text-guided attention mechanism that leverages the spoken content of salespeople to guide the model to focus toward intended products, emphasizing their salience over cluttered background products. Second, a long-range spatiotemporal graph network is further designed to achieve both instance-level interaction and frame-level matching, solving the misalignment caused by video-image heterogeneity. Third, we propose a multi-modal hard example mining, assisting the model in distinguishing highly similar products with fine-grained features across the video-image-text domain. Through extensive quantitative and qualitative experiments, we demonstrate the superior performance of our proposed SGMN model, surpassing the state-of-the-art methods by a substantial margin. The code and models will be public soon.

## CCS CONCEPTS

• **Information systems → Multimedia and multimodal retrieval**; • **Computing methodologies → Visual content-based indexing and retrieval**.

## KEYWORDS

Livestreaming Product Retrieval, Multi-modality, Graph learning

## 1 INTRODUCTION

Benefiting from the convenience of e-commerce, shopping online become increasingly popular in recent years. The livestreaming product retrieval (LPR) plays a crucial role in accurately matching products shown in live clips with those available in online

*MM '24, 28 October - 1 November 2024, Melbourne, Australia*
© 2024 Copyright held by the owner/author(s). Publication rights licensed to ACM.
ACM ISBN 978-1-4503-XXXX-X/18/06
https://doi.org/XXXXXXX.XXXXXXX

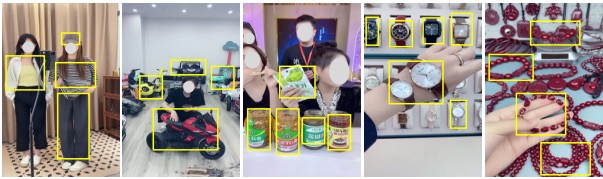

(a) Cluttered and similar background products.

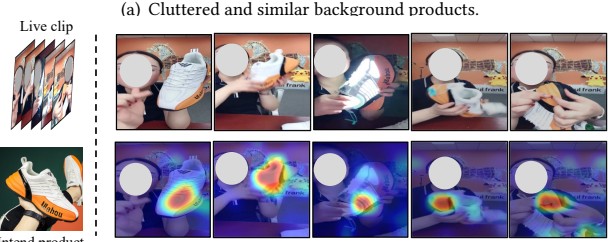

(b) Intended product retrieval with a large appearance variations.

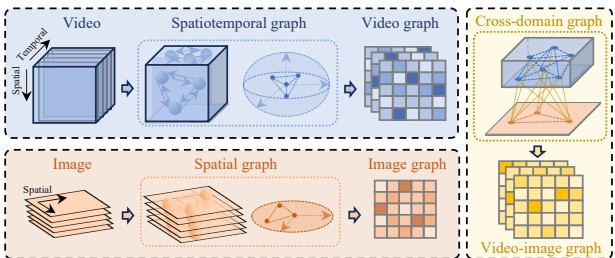

(c) Intra-domain and inter-domain spatiotemporal graphs.

**Figure 1: Representative examples of cluttered and similar background products (a) or large appearance variations such as occlusion, motion, and illumination (b). The intra-domain and inter-domain graphs (c) between videos and images to enhance the spatiotemporal frame-level interaction.**

shops, ensuring a seamless shopping experience and excellent marketing ability [11, 13, 25]. Even though various customer-to-shop retrieval methods have great progress [6, 12, 14], the cross-domain livestreaming-to-shop problem in LPR has few studies yet. Some unresolved challenges are still rooted in real-world applications.

First of all, accurately discerning the intended products offered by salespersons during livestreaming is profoundly challenging. It is common practice for salespersons to showcase multiple products but focus on promoting a specific product at a given time, defined as intended product. Some methods have attempted to incorporate an additional product detection module [12, 46], but they will incur high annotation costs and computational complexity, and also fail to completely eliminate ambiguity of intent. Sometimes, the detection of multiple foreground boxes introduces more uncertainty and erroneous guidance, as shown in Fig. 1(a).

Secondly, the heterogeneity between realistic livestreaming videos and product images further exacerbates the difficulty of cross-domain retrieval. As depicted in Fig. 1(b), livestreaming introduces variations in viewpoint, illumination, and occlusions, resulting in

significant appearance disparities between products in livestreaming and in online store. It is very common in live scenarios, where methods focused on the instance-level matching of entire clips will fail [6, 12, 44]. Coarse-grained instance-level matching makes it hard to track spatial structural deformations. The inevitable motion and occlusions will render products visually obscured or degraded during specific time slots, making frame-level sequential matching with temporal consistency necessary. Moreover, Large domain discrepancy leads to misalignment of intra-domain features. Existing cross-domain retrieval methods [6, 8, 24, 30, 41] often treat different domains equally while ignoring spatiotemporal relations within and between domains, further reducing retrieval accuracy.

Another dilemma in real-world applications lies in the abundance of highly similar products in online stores, necessitating that models excel at learning fine-grained representations and discriminating subtle distinctions among similar-looking products. As shown in Fig. 1(a), there exists a plethora of indistinguishable bracelets. Some methods based on pre-trained CLIP attempt to learn classifiable visual embeddings from large-scale video-text embeddings [4, 8, 21, 35]. However, generic features often fail to capture subtle product characteristics. This deficiency contributes to the suboptimal performance of the latest method, RICE [44], particularly in the crucial first-ranking metric (R@1).

To tackle the real-world challenges mentioned above, we propose the Spatiotemporal Graphing Multi-modal Network (SGMN) for LPR. Our model consists of three key components: 1) We leverage the verbal explanations provided by salespersons in livestreams, which often contain explicit information about the intended products. The texts from Automatic Speech Recognition (ASR) transcriptions and image titles guide the model to focus on products highly relevant to the verbal context, mitigating distractions from background clutter items. 2) We design a Graph-based Cross-domain Interaction (GCI) module to capture spatiotemporal correlations between videos and images. It is the first exploration of using sequence-to-sequence graph learning to model and enhance cross-domain temporal consistency and spatial correlation. As shown in Fig. 1(c), we establish the intra-domain and inter-domain connection graphs between video and image synchronously. Benefiting from the frame-level connectivity, the model can still accurately localize regions of intended products that encounter appearance distortion due to occlusion, motion, and brightness changes in Fig. 1(b). 3) The Selective Multi-modal Fusion (SMF) module is proposed to assist the model in distinguishing highly similar products. This mechanism selects the top-K hard examples in global ranking, and then fuses their visual and textual representations for implicitly recalibrating ranking and discerning semantic heterogeneity. The fine-grained alignment across the video-image-text domain demonstrates robust potential. Notably, our method adheres to the principle of independently encoding multi-modal inputs during inference, striking a balance between efficiency and accuracy. Our contributions can be summarized as follows:

- We introduce the textual information in the detector-free retrieval framework to guide model attention on the intended product. The comprehensive yet unified representation learned from the cross-modal alignment space of video-image-text solves the practical dilemma of LPR.

- The spatiotemporal graph learning is first explored for capturing sequential relations, alleviating inter-domain misalignments in both spatial and temporal dimensions.
- We selectively fuse multi-modal features to enhance fine-grained representations of hard samples, distinguishing products with subtle visual differences.
- Our method achieves the best performance on the large-scale benchmark dataset. Extensive quantitative and qualitative experiments prove the superiority of SGMN.

## 2 RELATED WORKS

### 2.1 Livestreaming Product Retrieval

Before the livestreaming became popular, researchers paid more attention to the customer-to-shop clothes retrieval tasks, such as fashion retrieval [10, 11, 13, 15, 25]. Motivated by video-to-shop retrieval, some works are dedicated to providing pair-matching solutions between image and video features, such as DPRNet [46] and SEAM [12]. Such methods adopt a two-stage framework combining detection and retrieval, localizing the products in the video before performing the global similarity match. AsymNet [6] uses a single-stage network that removes object detection to reduce model complexity. RICE [44] proposes a single-stage network without object detection to reduce model complexity. However, these methods have yet to effectively leverage the textual modality to identify the intended products. In our work, we make the first attempt at enhancing visual representations of intended products using the textual features in a one-stage framework.

### 2.2 Cross-domain Retrieval and Interaction

Existing cross-domain retrieval methods usually learn unified representations across different domains. Some methods feed concatenated visual and textual features into classifiers to predict image-to-text matching performance [4, 21, 34, 35]. CLIP2Video [8] transfers knowledge from pretrained CLIP to learn video-to-text features. Recent methods with high inference performance use independent encoders to extract global features and compute cross-domain similarity [28, 30, 32, 36, 48]. Methods for cross-domain interaction are also explored. SCAN [19] uses an attention mechanism to fuse across domains, and IMRAM [2] applies an iterative network for interaction. The multi-grained matching mechanism[30] shows limited performance in capturing long-range temporal dependency. Our work employs a spatiotemporal graph to enhance the cross-domain consistency across the video-image-text domain.

### 2.3 Reason Graph Learning

The graph reasoning network (GCN) has proven effective due to the powerful expressive capability of graph structures[3, 38]. In cross-domain retrieval tasks, using graph reasoning networks [16, 18, 47] to learn the relationship between different domains is a common practice. VSRN [20] models the local visual features of key objects semantically. CGMN [5] and GSMN [22] learn image-text matching using a cross-domain graph matching network. DSRAN [40] adopts the graph attention. Wang *et al.* used a coarse-to-Fine graph network for video-text retrieval [39]. The HREM [9] learns semantic relationships among image and text via a hierarchical graph relation model. These methods have shown the advantages of graph learning in modeling spatial correlation, but its potential

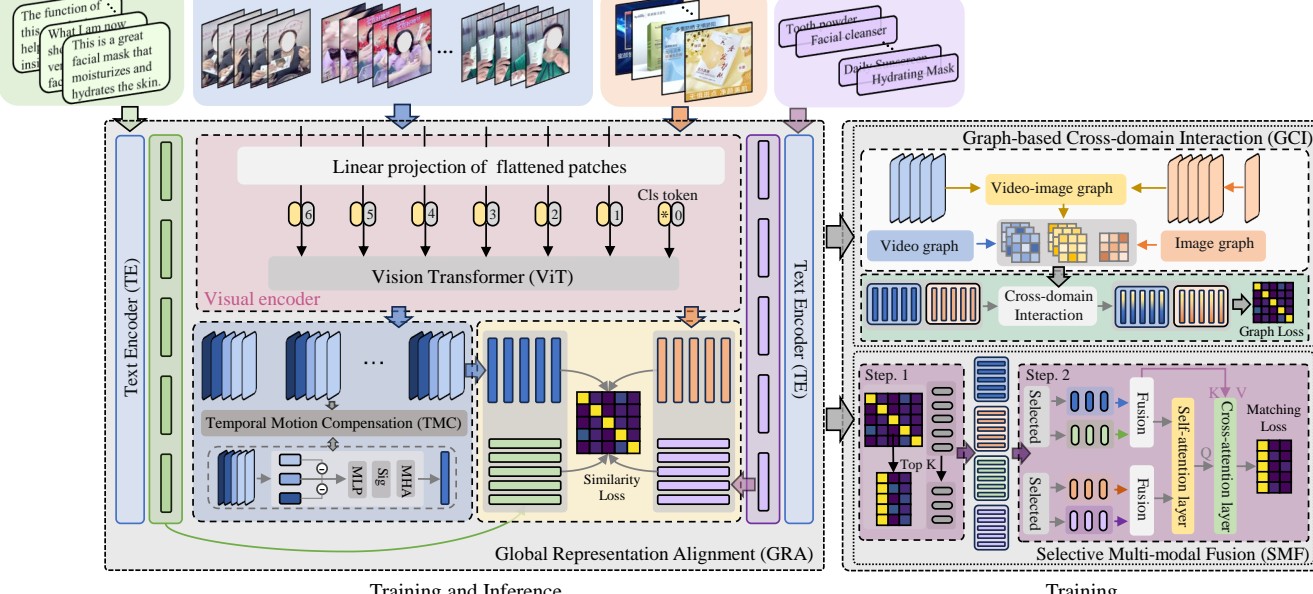

Figure 2: The architecture of the proposed SGMN. The inputs are live video clips, text from video ASR, product images, and product titles. Paired image-video and ASR-title representations in GRA module are independently encoded and weighted for global similarity. The GCI module (Top right) constructs the video graph, image graph, and video-image graph for cross-domain spatiotemporal relation learning. The SMF module (Bottom Right) selects hard examples and fuses multi-modal features for distinguishing mining. Only the GRA module is used for inference, while the GCI and SMF modules are applied for training.

in capturing temporal information within video sequences has yet to be explored. Our work has made preliminary attempts and validations in spatiotemporal sequence-level graphing.

## 3 METHOD

Our SGMN consists of three parts: the Global Representation Alignment (GRA), the Graph-based Cross-domain Interaction (GCI), and the Selective Multi-modal Fusion (SMF), as illustrated in Fig. 2. Further implementation details are elucidated in subsequent sections.

### 3.1 Global Representation Alignment

**Image Encoder.** We use the pretrained ViT-B/32 model from CLIP [29] as the image encoder. We define a batch of $N$ images as $I \in \mathbb{R}^{N \times H \times W}$, and the encoders extract non-overlapping image patches with the size of $p \times p$. Then we perform a linear projection to project these flattened patches into 1D markers as shown in Fig. 2. We then use the Multi-Head Attention (MHA) mechanism to interact with each patch of the image for the global aggregated features $I_{cls}$ and patch-level hidden embedding $I_{hid}$:

$$\{I_{cls}, I_{hid}\} = \mathcal{F}_{\text{MHA}}(Q_I, \mathcal{K}_I, \mathcal{V}_I), \quad (1)$$

where $Q$, $\mathcal{K}$, and $\mathcal{V}$ represent the query, key, and value. The $I_{cls} \in \mathbb{R}^{N \times D}$ and $D$ is the embedding dimension.

**Video Encoder.** The live streams are divided into clips $V \in \mathbb{R}^{N \times L \times H \times W}$, where $L$ is the length of video frames. We share parameters between the image encoder and the video encoder for global representation alignment, so the sequential features of video clips are encoded as:

$$\{V_{cls}, V_{hid}\} = \mathcal{F}_{\text{MHA}}(Q_V, \mathcal{K}_V, \mathcal{V}_V). \quad (2)$$

We also adopt the cls token $V_{cls} \in \mathbb{R}^{N \times L \times D}$ as the global representation of video clips. To enhance inter-frame alignment inside videos

and extract correlations within each frame, we use a Temporal Motion Compensation (TMC) module. TMC adds the displacement of inter-frame actions to the video sequence to simulate motion changes as $\Delta f_t = f^t - f^{t-1}$, where $f^{t-1}$ and $f^t$ represent two adjacent frames. We then encode the motion of consecutive frames and insert them as a motion compensation token for enhancing differential-level attention. The global video representation after motion-compensated enhancement is:

$$V_{visual} = \mathcal{F}_{\text{TMC}}(V_{cls} \mid \{\Delta f_1, \Delta f_2, \ldots, \Delta f_t\}). \quad (3)$$

**Text Encoder.** ChineseCLIP [43] is used to extract initial textual representations for video automatic speech recognition (ASR) $T_{asr}$ and image titles $T_{item}$. To keep the model lightweight, we pretrained the ChineseCLIP model and fixed the parameters in model training. Since the speech information from the salesperson always contains redundant product-independent information in the real-world livestreaming, we add a filter layer $\mathcal{F}_{\text{filter}}$ to eliminate irrelevant units while retaining key information.

$$V_{text} = \mathcal{F}_{\text{filter}}(\mathcal{F}_{\text{clip}}(T_{asr})),$$
$$I_{text} = \mathcal{F}_{\text{filter}}(\mathcal{F}_{\text{clip}}(T_{item})), \quad (4)$$

where the $\mathcal{F}_{\text{clip}}(\cdot)$ is the function of ChineseCLIP.

**Similarity Loss.** We use the triplet loss $\mathcal{T}(\cdot)$ [7] to measure the similarity of visual and textual embedding. The detailed implementation can be found in the supplementary. Therefore, the optimization goal of GRA is defined as the similarity loss $\mathcal{L}_s$ between video-to-image and text-to-text representations, and $\lambda$ is defined as the loss weighting factor.

$$\mathcal{L}_s = \mathcal{T}(V_{visual}, I_{visual}) + \lambda \mathcal{T}(V_{text}, I_{text}). \quad (5)$$

## 3.2 Graph-based Cross-domain Interaction

**Graphs Reconstruction.** We build a video graph, image graph, and video-image graph using global representations, as shown in Fig. 2. These spatiotemporal graphs can capture temporal consistency and spatial relations synchronously.

First, for a batch with $N$ video-image pairs, the sequence length of the video is $L$, so we can obtain the global video features as $S_V = V_{visual} \in \mathbb{R}^{N \times L \times D}$. Then we stack images into sequences of the same length as video clips for the sequence-to-sequence alignment, and the stacked global image features as $S_I = Stack(I_{visual}) \in \mathbb{R}^{N \times L \times D}$. The construction of the inter-domain spatiotemporal graph is shown in Fig. 3. The video-to-image similarity matrix $\mathcal{H}_{V2I} \in \mathbb{R}^{N \times L \times L}$ is represented as:

$$\mathcal{H}_{V2I} = S_V S_I^T. \tag{6}$$

However, fully connecting each <frame, image> pair while ignoring the relative importance is suboptimal [33]. Thus we define two intra-domain relation matrices and two inter-domain relation matrices, which are video spatiotemporal graph $G_{V2V}$, image spatial graph $G_{I2I}$ and video-image cross-domain graph $G_{V2I}$ and $G_{I2V}$. Then we build the entire spatiotemporal graph with intra-domain relations and cross-domain association as follows:

$$G = \begin{bmatrix} G_{I2I} & G_{I2V} \\ G_{V2I} & G_{V2V} \end{bmatrix} \tag{7}$$

Given $N$ image-video pairs, the size of four relation matrices are $\{G_{V2V}, G_{V2I}, G_{I2I}, G_{I2V}\} \in \mathbb{R}^{N \times L \times L}$. The matrices $G_{V2V}$ and $G_{I2I}$ represent the instance-level correlation of images or videos, while the matrices $G_{V2I}$ and $G_{I2V}$ represents the frame-level correlation between each image and each video frame. All nodes and edges in graphs independently and jointly represent objects and relations of multiple domains. Then we refer to the frame-by-frame matched scores to filter out irrelevant connectivity and define two semantic properties: *Connection* and *Relevance*.

**i) Connection.** To build an efficient correlation graph and preserve useful spatiotemporal connections, we follow an advanced fine-grained matching scheme [19]. The connection relationships between each video frame and the image depend on the matching relationships of multiple nodes.

As shown in Fig. 3, given the frame-image similarity matrix, each column vector represents the matching relationships between each video frame and a certain image. We first compute the mean of the similarity and then measure the top-K matching pairs. We calculate the difference between the mean value and top-K values as:

$$C_i = \text{Mean}(\mathcal{H}_{V2I}^i) - \text{topK}(\mathcal{H}_{V2I}^i, K), \tag{8}$$

where $i \in [0, L)$ and $\mathcal{H}_{V2I}^i$ is the $i$-th column vector and $K$ is set as $L/2$. Finally, the connection property between the frame and the image is determined based on the logical values of $C$. The $\mathcal{M} \in \mathbb{R}^{N \times L \times L}$ is the connection mask:

$$\begin{aligned} \mathcal{M}_{i,j} = 1 \text{ if } C_{i,j} >= 0, \\ \mathcal{M}_{i,j} = 0 \text{ if } C_{i,j} < 0, \end{aligned} \tag{9}$$

**ii) Relevance.** Due to the heterogeneity between videos and images, simply using global embeddings for cross-modal relevance learning is insufficient. So we continue to employ fine-grained matching based on multiple nodes to narrow the cross-domain differences.

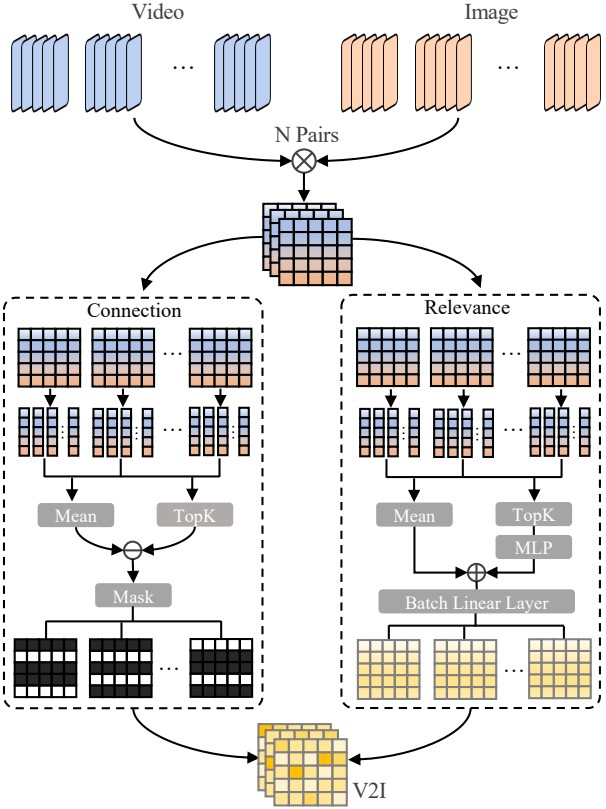

**Figure 3: Graph construction of a cross-domain connection between video and image sequences within a batch.**

For the video-to-image associations, we first extract the column vectors of $\mathcal{H}_{V2I}^i$ and then simultaneously compute the mean value and remap the top-K matching scores:

$$\mathcal{A}_i = \text{Mean}(\mathcal{H}_{V2I}^i) + \mathcal{F}_{\text{MLP}}(\text{topK}(\mathcal{H}_{V2I}^i, K)), \tag{10}$$

where $i \in [0, L)$ is the column index and $K = L$. Since the relevance learning for different samples within a batch is independent, we use an additional batch linear layer to encourage interactions between different samples and further expand the matching receptive field.

$$\mathcal{R}_i = \mathcal{F}_{\text{BLL}}(\mathcal{A}_i), \tag{11}$$

where the $\mathcal{F}_{\text{BLL}}(\cdot)$ includes an ReLU activation layer and two linear layers. Then we obtain the relevance matrix $\mathcal{R}$ for all pairs of video-to-image instances.

Therefore, the cross-domain graph can be constructed from the connection matrix $C$ and relevance matrix $\mathcal{R}$ as:

$$G_{V2I} = \mathcal{R} \otimes C. \tag{12}$$

**Cross-domain Interaction.** As shown in Fig. 4, we design two attention branches to enhance the visual representation. The cross-domain connections in GCI represent the interaction of intra-domain and inter-domain features. We use the constructed graph $G_{I2V}$ and $G_{V2I}$ to replace the mask in the MHA function to guide the model attention to local areas with high relevance as:

$$\begin{aligned} I_m = \mathcal{F}_{\text{LN}}(\mathcal{F}_{\text{MHA}}(Q_I, \mathcal{K}_V, \mathcal{V}_V \mid G_{I2V})), \\ V_m = \mathcal{F}_{\text{LN}}((\mathcal{F}_{\text{MHA}}(Q_V, \mathcal{K}_I, \mathcal{V}_I \mid G_{V2I})), \end{aligned} \tag{13}$$

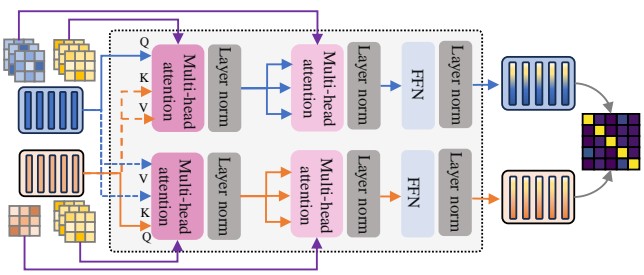

**Figure 4: Details of the cross-domain interaction module.**

where $\mathcal{F}_{\text{LN}}(\cdot)$ is the layer norm function. Then we take the graph-enhanced feature $I_m$ and $V_m$ through the MHA module under the guidance of the intra-modal graphs $G_{I2I}$ and $G_{V2V}$ to obtain the graph-enhanced visual representation $I_g$ and $V_g$, and $\mathcal{F}_{\text{FFN}}(\cdot)$ is the function of a feed-forward network.

$$
\begin{aligned}
I_g &= \mathcal{F}_{\text{FFN}}(\mathcal{F}_{\text{MHA}}(Q_{I_m}, \mathcal{K}_{I_m}, \mathcal{V}_{I_m} \mid G_{I2I})), \\
V_g &= \mathcal{F}_{\text{FFN}}(\mathcal{F}_{\text{MHA}}(Q_{V_m}, \mathcal{K}_{V_m}, \mathcal{V}_{V_m} \mid G_{V2V})).
\end{aligned}
\tag{14}
$$

**Graph Loss.** We first use the triplet loss $\mathcal{T}(\cdot)$ to compute the cosine similarity of features augmented by spatiotemporal graphs, jointly optimizing the parameters of the image, video, and video-image graphs. In addition, the cross loss between the enhanced and the original embedding is additionally calculated to narrow the feature domain difference, and the graph loss is:

$$
\mathcal{L}_{gr} = \mathcal{T}(V_g, I_g) + \mathcal{T}(V_{visual}, I_g) + \mathcal{T}(V_g, I_{visual}).
\tag{15}
$$

We use the KL-divergence loss [37] function $\mathcal{K}(\cdot)$ to minimize the difference between the distribution of calibrated cross-domain correlations and the global embedding.

$$
\mathcal{L}_{kl} = \mathcal{K}(G_{V2I}, \mathcal{H}_{V2I}) + \mathcal{K}(G_{I2V}, \mathcal{H}_{I2V}).
\tag{16}
$$

And the total loss in GCI is defined as $\mathcal{L}_g = \mathcal{L}_{gr} + \mathcal{L}_{kl}$.

### 3.3 Selective Multi-modal Fusion

To improve the discrimination of semantically related and visually similar products, we further perform the hard negative mining operation using the multi-modal features. First, we compute a similarity matrix with visual and textual representations in the GRA module to select hard examples as follows:

$$
M_{sim} = V_{visual} \cdot I_{visual}^{\text{T}} + \alpha V_{text} \cdot I_{text}^{\text{T}},
\tag{17}
$$

then we select top $K$ samples ranked in $M_{sim}$ as the hard samples for further matching. The $K$ is set to 4. The visual and textual features of these samples are selected as:

$$
\begin{aligned}
ind &= \text{topK}(M_{sim}, K), \\
\hat{V}_{visual}, \hat{I}_{visual} &= V_{visual}[ind], I_{visual}[ind], \\
\hat{V}_{text}, \hat{I}_{text} &= V_{text}[ind], I_{text}[ind].
\end{aligned}
\tag{18}
$$

Besides, we fuse the textual feature and visual representations of selected samples using a fusion layer $\mathcal{F}_{\text{fusion}}(\cdot)$. The multi-modal features of video and image are learned.

$$
\hat{V} = \mathcal{F}_{\text{fusion}}(\hat{V}_{visual}, \hat{V}_{text}), \quad \hat{I} = \mathcal{F}_{\text{fusion}}(\hat{I}_{visual}, \hat{I}_{text}).
\tag{19}
$$

The fused features $\hat{V}$ and $\hat{I}$ will pass through a perceptual module consisting of a self-attention layer and a cross-attention layer $\mathcal{F}_{cross}$

to obtain cross-domain features:

$$
V_{cross} = \mathcal{F}_{\text{cross}}(Q_{\hat{I}}, \mathcal{K}_{\hat{V}}, \mathcal{V}_{\hat{V}}),
\tag{20}
$$

Then we use the symmetric cross-entropy loss $\mathcal{E}(\cdot)$ [8] to calculate the matching loss in hard negative mining.

$$
\mathcal{L}_m = \mathcal{E}(\text{AvgPool}(V_{cross})).
\tag{21}
$$

### 3.4 Optimization Target

To learn a comprehensive representation from the video-image-text alignment space, the total loss is optimized as:

$$
\mathcal{L}_{total} = \mathcal{L}_s + \beta_1 \mathcal{L}_g + \beta_2 \mathcal{L}_m.
\tag{22}
$$

Mentioned that the model only uses the independent embedding in GRA to calculate the global similarity in the inference stage, having high matching accuracy and efficiency.

### 3.5 Model Inference

During the inference stage, only the GRA module is utilized to independently encode the global visual and textual representations of the input video and the product gallery images. The matching scores $\mathcal{S}$ are calculated as a weighted similarity of the embeddings from the visual and text domain.

$$
\mathcal{S} = V_{visual} I_{visual}^T + \lambda V_{text} I_{text}^T.
\tag{23}
$$

## 4 EXPERIMENTS

### 4.1 Experimental Setup

**Datasets.** We conduct experiments on the publicly available dataset LPR4M [44] and MovingFashion (MF) [12]. The LPR4M dataset is the largest publicly available multi-modal dataset that covers 34 categories and comprises three modalities (image, video, and text). It includes 3,955,181 pairs of video-images for training and 20,079 pairs for testing, encompassing a diverse range of scenes. The MF dataset contains only one category of fashion clothing with two modalities (image, video), comprising 15,045 pairs in the training set and 1,341 pairs in the test set. These well-stocked commerce datasets facilitate a fair evaluation of our method in real-world scenarios. More dataset analyses are available in the supplements.

**Implementation Details.** We initialize image and video encoders using the pretrained ViT-B/32 model from CLIP [29] with a feature dimension of 512. The text encoder is initialized by the pretrained RoBERTa-wwm-Base model from ChineseCLIP [43]. For video preprocessing, we extract 10 evenly spaced frames from each clip. Each frame in videos has a probability of 0.5 to be randomly masked for data augmentation, and the masking percentage ranges from 0 to 0.9. We set the weight factor $\lambda = 0.5$ in Eq. 5 and Eq. 23. The margin in triplet loss is set to 0.2 for proper discrimination. In Eq. 8, top-K scores serve as the relevance threshold, set to $L/2$ to balance performance and complexity. In Eq. 17, higher $K$ implies more memory (M) costs, so we set $K=4$. The weight factors are $\beta_1 = 0.7$ and $\beta_2 = 0.3$ in Eq. 22. In model optimization, we use the Adam [17] optimizer with a batch size of 256. The learning rate is $3 \times 10^{-4}$, which decays following the cosine schedule [27]. Our models are trained on 8 NVIDIA Tesla V100 GPUs. Following the standard retrieval task [31, 45], recall at rank K (R@K) is adopted as the metric to evaluate the performance quantitatively.

Table 1: Performance comparison on the LPR4M dataset.

| Methods | R@1 | R@5 | R@10 | R@mean |
|---|---|---|---|---|
| FashionNet [25] | 13.4 | 33.8 | 50.4 | 32.5 |
| AsymNet [6] | 22.0 | 46.7 | 63.8 | 44.2 |
| SEAM [12] | 23.3 | 49.5 | 61.4 | 44.7 |
| TimeSFormer [1] | 28.6 | 56.8 | 69.0 | 51.5 |
| NVAN [23] | 21.4 | 45.2 | 62.7 | 43.1 |
| SwinB [26] | 29.1 | 60.1 | 73.9 | 54.4 |
| RICE [44] | 33.0 | 65.5 | 77.3 | 58.6 |
| SGMN (w/o TE) | 38.7 | 66.5 | 76.2 | 60.5 |
| **SGMN (Ours)** | **43.4** | **68.9** | **79.2** | **63.8** |

Table 2: Performance comparison on the MF dataset.

| Methods | R@1 | R@5 | R@10 | R@mean |
|---|---|---|---|---|
| NVAN [23] | 38.0 | 62.0 | 70.0 | 56.7 |
| MGH [42] | 40.0 | 59.0 | 66.0 | 55.0 |
| AsymNet [6] | 42.0 | 73.0 | 86.0 | 67.0 |
| SEAM [12] | 49.0 | 80.0 | 89.0 | 72.7 |
| RICE [44] | 76.1 | 89.7 | 92.6 | 86.1 |
| SGMN (w/o Finetune) | 43.1 | 63.5 | 70.7 | 59.1 |
| **SGMN (Ours)** | **77.8** | **90.3** | **92.7** | **86.9** |

## 4.2 Comparison with Other Methods

We quantitatively compare the proposed SGMN with the existing methods on the LPR4M dataset and MF dataset, and the results are shown in Tab. 1 and Tab. 2. On the large-scale dataset LPR4M, the retrieval performance of our method outperforms the state-of-the-art RICE [44] by 10.4% in R@1, 3.4% in R@5, and 1.9% in R@10. Existing methods exhibit limited performance on LPR4M with rich categories and diverse scenes. We also verified the performance of SGMN without using textual features (w/o TE). Our SGMN can still achieve a performance gain of 5.7% on R@1 without the textual guidance to the salient regions (38.7% vs 33.0%).

Since the MF dataset does not have text modality, we do not fuse the textual information in training. Efficient global cross-domain alignment still allows SGMN to maintain the best performance with significant gains. Compared with the official method of MF dataset SEAM, our method improves R@1 by 28.8% in R@1. Although the MF testset contains only 1,342 videos, our approach still gains a 1.7% advantage over the highest RICE method in the crucial R@1 metric, signifying our superiority in extracting fine-grained discriminative features. We also verified the model trained on LPR4M without finetuning on MF, and the results show that even if the model has not seen the MF data, it can still surpass most trained models, demonstrating robust zero-shot fitting and generalization capability. To summarize, our method benefits from cross-domain connections and fine-grained matching, showing greater superiority on both simple and diverse datasets, particularly in the crucial R@1 metric.

## 4.3 Qualitative Results

**Retrieval Results.** We randomly select several video sequences from MF dataset and LPR4M dataset, and the top-3 shop images are shown in Fig. 5. It can be seen that even if the live scene is filled with numerous products of the same type, our SGMN can still find

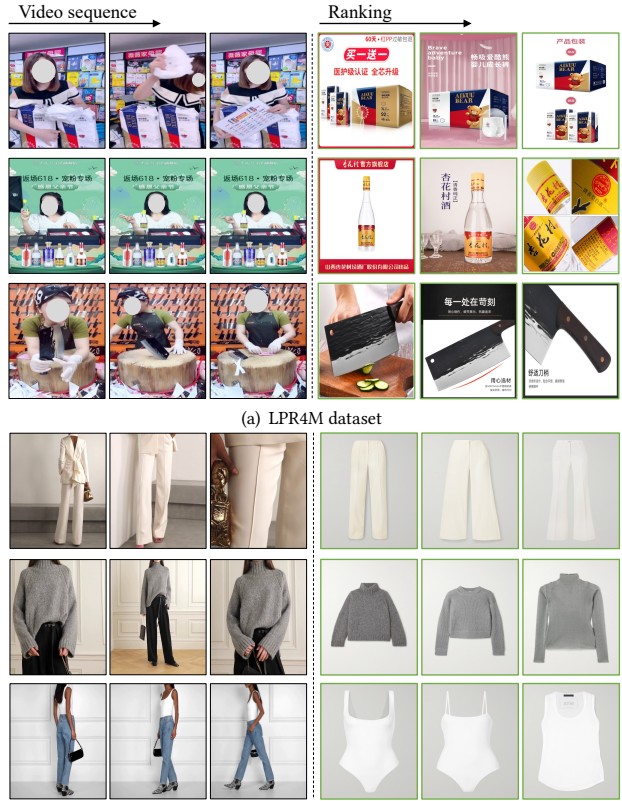

(a) LPR4M dataset

(b) MF dataset

Figure 5: Ranking results of representative products in LPR.

Table 3: Comparison of inference time of different methods.

| Method | AsymNet | SEAM | TimeSFormer | RICE | SGMN |
|---|---|---|---|---|---|
| Time (ms) | 910.4 | 622.6 | 163.2 | 24.0 | 24.3 |
| Params (M) | 295 | 52 | 185 | 158 | 174 |
| R@1 (%) | 22.0 | 23.3 | 28.6 | 33.0 | **43.4** |

the best matched product. Besids, our model performs excellently in distinguishing highly similar clothes in the MF dataset. Even though the intended products encounter appearance distortion due to occlusion, motion, scaling, or illumination variations, our method can still accurately retrieve the correct products.

**Inference Time.** We show the model inference time in Tab. 3 to verify the efficiency of our one-stage strategy. Experiments for different methods were conducted following their open-source setting and replicated on the Tesla T4 GPU. Compared to two-stage methods like AsymNet and SEAM, which require additional object detection, our SGMN achieves over 30 times faster speed. Even with the TE module, our SGMN demonstrates nearly identical runtime to RICE using the same CLIP model. Benefiting from globally independent encoding and parameter sharing, our SGMN balances well between model complexity and performance.

**Generalization and Robustness.** We carried out experiments on different video variations of the LPR4M dataset and the results are in Tab. 4. $RICE_{patch}$ is a one-stage algorithm, while $RICE_{box}$ is a two-stage algorithm manually incorporating detected boxes. Data is divided into small, medium, and large subsets with different product scales, visible durations, and product numbers. Our SGMN

**Table 4: The R@1 performance of various methods is evaluated on the LPR4M dataset across subsets categorized by video variations, including product scale, visible duration, and number of products. The best performance for each subset is highlighted.**

| Methods | overall | scale | | | visible duration | | | number of product | | |
|---|---|---|---|---|---|---|---|---|---|---|
| | | small | medium | large | short | medium | long | abundant | medium | few |
| $\text{RICE}_{patch}$ | 31.2 | 28.9 | 37.0 | 32.7 | 28.1 | 32.9 | 39.6 | 21.0 | 34.6 | 31.5 |
| $\text{RICE}_{box}$ | 33.0 | 32.7 | 39.0 | 33.8 | 29.6 | 34.8 | 42.0 | 17.6 | 31.9 | 33.4 |
| **SGMN (Ours)** | **41.5** | **38.3** | **43.0** | **54.9** | **36.6** | **43.6** | **54.5** | **24.7** | **32.3** | **42.9** |

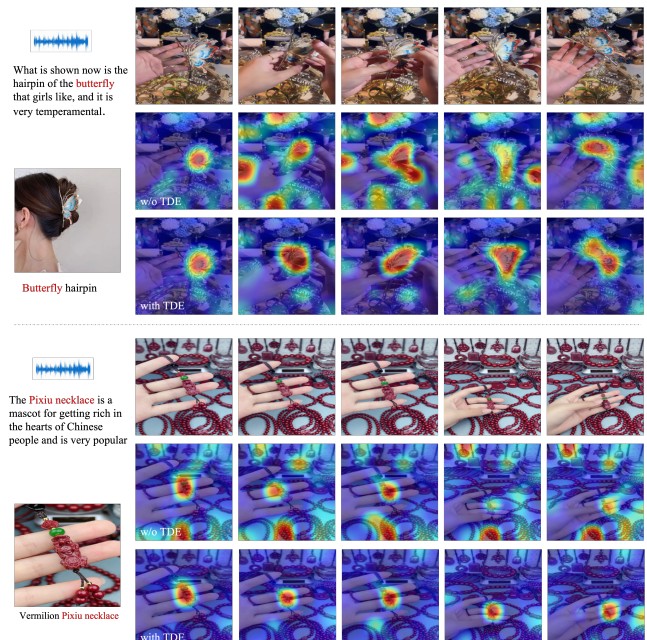

**Figure 6: Attention visualization with and without (w/o) TE modules. Benefiting from the embedding and guidance of text information, the model tends to focus on keyword-related products in the cluttered background. The keywords 'butterfly' and 'Pixiu necklace' that co-exist in the video text and product items correct the model attention efficiently.**

consistently outperforms RICE by a significant margin across various subsets, demonstrating robustness in real-world scenarios.

## 4.4 Ablation Study

**Text Attention Analysis.** Embedding text from video ASR and product items can guide the model to focus on the intended item in the cluttered background at a low cost. Therefore, we visualized the attention of the video encoder in GRA equipped with or without (w/o) the TE module. As shown in Fig. 6, the method with TE is superior to that w/o TE in accurately retrieving the intended product. For example, in the live clip of selling hairpins, multiple similar hairpins appear in view simultaneously, interfering with the identification of intended products. In this case, the keyword 'butterfly' mentioned by the live salesperson played a key role, significantly increasing the probability of matching products that contain both 'butterfly' and 'hairpin' in the title. Similarly, there are a large number of extremely similar products in the vermilion necklace category gallery, and their visual differences are almost negligible in the live domain, where the resolution is degraded. Fortunately, the assistance of text information solves this dilemma very

**Table 5: Hyperparameters analysis of loss function.**

| $\beta_1$ | 0 | 0.2 | 0.3 | 0.5 | 0.7 | 0.8 | 1 |
|---|---|---|---|---|---|---|---|
| $\beta_2$ | 1 | 0.8 | 0.7 | 0.5 | 0.3 | 0.2 | 0 |
| R@1 | 39.1 | 41.8 | 40.7 | 41.5 | **43.4** | 39.4 | 40.9 |

**Table 6: Performance of combining different relation graphs, including intra- and inter-domain graphs of video and image.**

| # | $G_{I2I}$ | $G_{V2V}$ | $G_{I2V+V2I}$ | R@1 | R@5 | R@10 |
|---|---|---|---|---|---|---|
| A | | | | 38.6 | 65.7 | 76.5 |
| B | ✓ | | | 38.8 | 65.9 | 77.0 |
| C | | ✓ | | 39.7 | 66.7 | 77.3 |
| D | | | ✓ | 40.1 | 67.0 | 77.8 |
| E | ✓ | ✓ | | 39.9 | 67.2 | 77.6 |
| F | ✓ | | ✓ | 40.5 | 67.5 | 78.1 |
| G | | ✓ | ✓ | 40.8 | 67.9 | 78.6 |
| H | ✓ | ✓ | ✓ | **43.4** | **68.9** | **79.2** |

**Table 7: Model component ablation on the LPR4M dataset.**

| # | TE | TMC | GCI | SMF | R@1 | R@5 | R@10 |
|---|---|---|---|---|---|---|---|
| A | | | | | 32.8 | 58.7 | 72.7 |
| B | ✓ | | | | 37.3 | 63.8 | 74.4 |
| C | ✓ | ✓ | | | 38.2 | 65.2 | 76.1 |
| D | ✓ | ✓ | ✓ | | 40.9 | 67.3 | 77.8 |
| E | ✓ | ✓ | ✓ | ✓ | **43.4** | **68.9** | **79.2** |

well. The keywords 'Pixiu' and 'necklace' have endowed the product with specificity, freeing the network from the interference of redundant products irrelevant to the keywords. These results show that our SMGN equipped with additional textual representations alleviates the difficulty of product-heavy recognition.

**Loss Function Analysis.** As shown in the Tab. 5, we analyze the weights of different components in the loss function. It can be observed that not using graph loss or hard example loss is unwise. A higher weight for the graph loss brings more performance gain, but it needs to be balanced with the SMF module. The frame-level temporal correlation complements high similarity discrimination, so we set $\beta_1$ and $\beta_2$ in Eq. 22 as 0.7 and 0.3 respectively.

**Different Spatiotemporal Graphs.** To analyze the performance of spatiotemporal graph learning in enhancing cross-domain interaction, we verified the performance using different relation graphs, and the results are shown in Tab. 6. The baseline uses the self-attention mechanism to interact with the initial features of videos and images. It can be seen that using graph learning to enhance intra-domain correlation is helpful because the methods using $G_{I2I}$ or $G_{V2V}$ graphs both have performance gains. Besides, constructing $G_{I2V}$ and $G_{V2I}$ graphs with cross-domain interactions has a higher gain in R@1 (40.1% vs 38.6%), indicating that the spatiotemporal

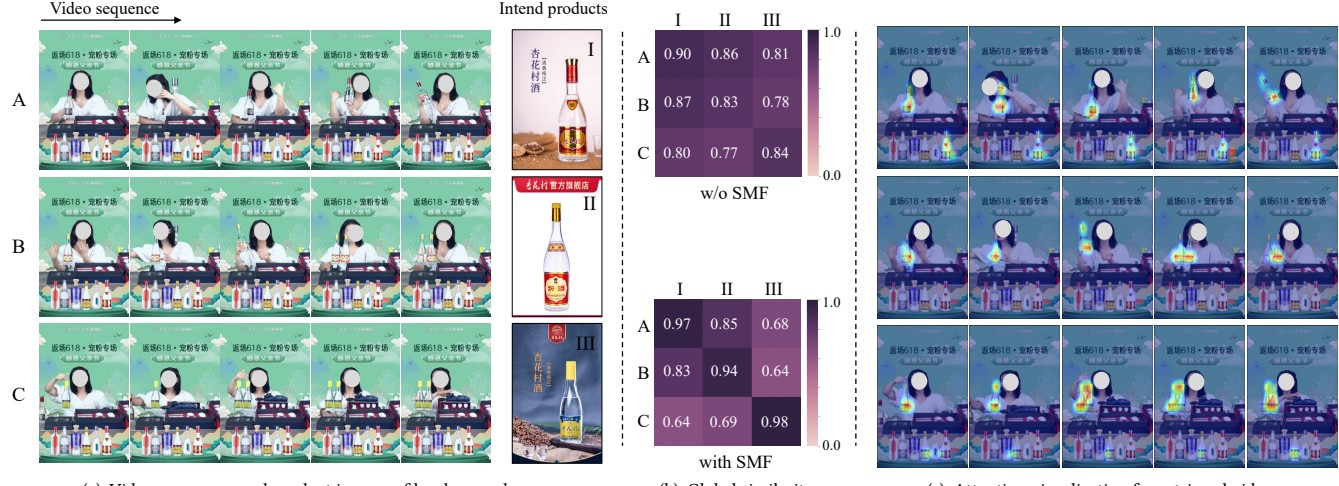

(a) Video sequences and product images of hard examples.  (b) Global similarity  (c) Attention visualization for retrieved videos

**Figure 7: Quantitative and qualitative comparison of our proposed SMF module in identifying multiple highly similar products. (a). Three similar-looking liquor video sequences (A, B, and C) and corresponding intended product images (I, II, and III). (b). Quantification of the global embedding similarity between the three videos and the corresponding product images using the model with or without the SMF module. (c). Attention visualization of our SGMN on encoded features for three video sequences.**

aligned domain is more conducive to learning efficient representations. Since products in the live domain often have appearance deformations, capturing enough fine-grained and long-range features to distinguish video-to-image pairs accurately is critical. The jointly modelled temporal consistency and spatial correlation enhance the global embedding of images and videos, achieving a 2.9% accuracy gain in R@1 (41.5% vs 38.6%).

**Identification of Similar Products.** In livestreaming e-commerce, similar products have incredibly similar appearance and visual features and often appear in the same field of view to facilitate sales, further exacerbating recognition accuracy. Therefore, to illustrate that our proposed SMF module strengthens the model in distinguishing hard negative samples, we selected several video sequences with top scores in the 'Liquor' category retrieval for quantitative and qualitative comparisons. As shown in Fig. 7(a), the product images of the three liquors have extremely slight visual differences, and their corresponding live video sequences are also easily confused. We quantify the similarity scores of the three videos (A, B, and C) and the corresponding product images (I, II, and III) with and without (w/o) the SMF module, and the results are shown in Fig. 7(b). It can be seen that the model without SMF has poor scoring discrimination for similar products, and the scores of the correct category and the wrong category are very close, which increases the probability of misclassification. On the contrary, the model that further performs hard example mining using SMF has higher discriminative power for similar products. Furthermore, we visualize the visual attention of our SGMN on three video sequences in Fig. 7(c). The mining of hard samples incorporating multi-modal features captures sufficient fine-grained features to recalibrate model attention on intended products accurately and efficiently.

**Model Component Ablation.** We analyze the performance of each component in our method on the LPR4M dataset and present the quantitative results in Tab. 7. The baseline is a GRA module that independently encodes visual representations. It can be seen

that each component we designed is effective. Among them, the TE introducing text attention and GCI learning spatiotemporal cross-domain interaction achieved higher performance gains, which are 4.5% and 2.7% in R@1, respectively. TMC enhances the intra-frame correlation of the global embedding of the video, and SMF further distinguishes similar samples, achieving R1 gains of 0.9% and 0.6%. Overall, the GCI module emphasizes frame-level fine-grained correlation over global coarse-grained alignment, while the SMF module achieves highly similar semantic discrimination at minimal cost. Our proposed solution aims to comprehensively optimize spatiotemporal consistent multi-modal representations, addressing the application dilemma of LPR step by step.

## 5 CONCLUSION

In this paper, we rethink the LPR task from a more macroscopic and practical point of view and propose a one-stage spatiotemporal graphing network to address the real-world dilemmas of LPR tasks. To the best of our knowledge, this is the first exploration of sequence-to-sequence graph learning in mitigating heterogeneity between videos and images. Existing methods are limited by their narrow focus on coarse-grained matches. But we progressively enhance the fine-grained discrimination by integrating *modality-level* text embeddings, *instance-level* similarity mining, and *frame-level* graph learning. Despite potential distortions in appearance within the complex livestreaming domain, our method adeptly tracks spatial deformations, ensuring precise location of the intended product. Moreover, we have fully harnessed the accessible textual information from live ASR transcripts and product titles, freeing the network from interference over cluttered background products. The further hard negative mining in video-image-text alignment domain improves the ability of our model to distinguish highly similar products. Extensive quantitative experiments show that our method well satisfies the demand for both local fine-grained attention and global spatiotemporal awareness in real-world scenarios.

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
