# OpenReview forum: "Spatiotemporal Graph Guided Multi-modal Network for Livestreaming Product Retrieval"
_acmmm.org/ACMMM/2024/Conference — MM2024 Poster_

### Official Review · Reviewer_H2j1 · 2024-05-22

**Rating:** 6
**Confidence:** 3

**Summary:**

To tackle the three challenges, i.e., the recognition of intended products from distractor products, the video-image heterogeneity and the confusing products with subtle visual nuances, in livestreaming product retrieval, this paper proposes a spatiotemporal graphing multi-model network. In the proposed network, a text-guided attention mechanism leverages the spoken content of salespeople to focus on intended products, a spatiotemporal graph network is designed to solving the video-image heterogeneity, and a multi-model hard example mining is proposed to distinguish highly similar products.

**Strengths:**

The novelty of this paper primarily reflect in two main aspects: (a) This paper leverages the textual modality to identify the intended products from distractor product, and (b) this paper uses the spatiotemporal graph learning to alleviate the inter-domain misalignment in both spatial and temporal dimensions.
Comprehensive experiments are conducted to prove the performance of the proposed method, including the quantitative comparison with other methods, the qualitative analysis of retrieval results, and the ablation study. The experimental results show that the proposed method outperforms the compared methods.
The paper is well wrote.

**Limitations:**

Some problems about experiments are listed:
How many images and videos are used in the training of the proposed method? Is the same images and videos used for the training of the compared methods?
Is the R@mean a popular metric in the livestreaming product retrieval, and how to compute this metric?
It is better to give more explanations about the data partitioning about product scale, visible duration, and number of products in table 4.

**Suitability:**

3

---

### Official Review · Reviewer_opQK · 2024-05-25

**Rating:** 3
**Confidence:** 4

**Summary:**

This paper proposes a novel approach to improve the retrieval of products in livestreaming contexts. The core of the method is the Spatiotemporal Graphing Multi-modal Network (SGMN), which integrates video, image, and text data to enhance the identification and differentiation of products. Experimental results on two public datasets have demonstrated the validity of the model.

**Strengths:**

1.The content of the article is relatively abundant.
2.The experiment is relatively adequate.

**Limitations:**

1.How does livestreaming product retrieval differ from traditional product retrieval and recommendation? Can't the original method be applied to the task? In live streaming, images of products and purchase links are usually given, would it be simpler and more effective to use images directly for retrieval?
2.The necessity of this task needs to be further emphasized, as well as practical application scenarios. To make the task easier to understand, a task definition is also needed.
3.Each module needs to be introduced with the reason for its introduction to enhance the logic of the article. For example, why create video graph, image graph, and video-image graph?
4.Many methodological and experimental details are not given in full, such as exactly how the TMC module is designed.
5.Several works in related fields should be cited, e.g.,
[1] Bi-directional Heterogeneous Graph Hashing towards Efficient Outfit Recommendation.
[2] Contrastive Collaborative Filtering for Cold-Start Item Recommendation.
6.A single Recall metric is not sufficient. In the retrieval domain, metrics such as accuracy, NDCG, etc., are usually also provided.

**Suitability:**

3

---

### Official Review · Reviewer_U8jr · 2024-05-25

**Rating:** 6
**Confidence:** 3

**Summary:**

The paper proposes an one-stage spatiotemporal graph network for livesreaming product retrieval, called SGMN. Using text information obtained from video ASR and product items to guide model focus on the intended product. Using sequence-to-sequence graph learning to mitigate heterogeneity between videos and images and progessively enhance the fine-grained matching ability. Hard negative minging further improve the fine-grained representations. Quantitative and qualitative experiments demonstrate the superior performance of the proposed SGMN model.

**Strengths:**

1. The proposed method achieves the best performance on the largescale benchmark dataset. Extensive quantitative and qualitative experiments prove the superiority of SGMN.
2. The visualization part for text guided attention and multi-modal hard negative minging is intersting and convincing.
3. Sufficient ablation experiments verified the effectiveness of the program design.

**Limitations:**

Since the R@mean metric of the LPR4M dataset is only 63.8, it suggests there is significant room for improvement. Please present some failed cases for analysis and provide an explanation of the limitations of this approach.

**Suitability:**

2

---

### Official Review · Reviewer_mv8P · 2024-05-26

**Rating:** 4
**Confidence:** 3

**Summary:**

This paper pays attention to the livestreaming product retrieval (LPR) problem. To provide a solution, a new method called Spatiotemporal Graphing Multi-modal Network (SGMN) is proposed. This method is applied to two publicly available datasets LPR4M and MF.

**Strengths:**

1. The topic is livestreaming product retrieval, which is definitely interesting and challenging. This paper not only give its insights of this task but also proposes a new method for it.
2. The proposed method SGMN is technically sound. Extensive experiments in section 4 and supplementary have shown its effectiveness with both qualitative and quantitative results.
3. The experimental design is convincing. And the experimental results are sufficient.
4. For good reproducibility, the code of the proposed method has been provided for review. Besides, this paper promises that the code and models will be public.

**Limitations:**

1. References to literature is inadequate and more closely related works should be included.

**Suitability:**

3

---

### Meta-Review · Area_Chair_j37j · 2024-07-03

**Recommendation:** Accept (Poster)
**Confidence:** 4

**Metareview:**

This paper presents a novel and promising approach to livestreaming product retrieval. The proposed SGMN method demonstrates strong performance on public datasets and is supported by comprehensive experiments. While there are some limitations in terms of literature coverage and experimental details, the overall contribution appears significant. Given the positive aspects highlighted by all reviewers and the addressed concerns in the rebuttal, I recommend acceptance for this paper. I strongly encourage the authors to address the noted limitations in the camera-ready version.